DATA RELEASE

# *De novo* chromosome-length assembly of the mule deer (*Odocoileus hemionus*) genome

Sydney Lamb[1,2,*,†], Adam M. Taylor[1,†], Tabitha A. Hughes[1], Brock R. McMillan[1], Randy T. Larsen[1], Ruqayya Khan[3], David Weisz[3], Olga Dudchenko[3,4], Erez Lieberman Aiden[3,4,5,6,7], Nathaniel B. Edelman[8] and Paul B. Frandsen[1,9,*]

1   Department of Plant and Wildlife Sciences, Brigham Young University, Provo, UT 84602, USA
2   Utah Division of Wildlife Resources, Salt Lake City, UT 84114, USA
3   The Center for Genome Architecture, Department of Molecular and Human Genetics, Baylor College of Medicine, Houston, TX 77030, USA
4   Center for Theoretical Biological Physics and Department of Computer Science, Rice University, Houston, TX 77030, USA
5   UWA School of Agriculture and Environment, The University of Western Australia, Crawley, WA 6009, Australia
6   Broad Institute of MIT and Harvard, Cambridge, MA 02139, USA
7   Shanghai Institute for Advanced Immunochemical Studies, ShanghaiTech, Pudong 201210, China
8   Yale Institute for Biospheric Studies, New Haven, CT 06520, USA
9   Data Science Lab, Office of the Chief Information Officer, Smithsonian Institution, Washington, DC 20002, USA

**Submitted:**   13 August 2021

\*   Corresponding authors. E-mail: slamb17@gmail.com; paul_frandsen@byu.edu

†   Contributed equally.

Preprint submitted at https://doi.org/10.1101/2021.08.12.456132

## ABSTRACT

The mule deer (*Odocoileus hemionus*) is an ungulate species that is distributed in a range from western Canada to central Mexico. Mule deer are an essential source of food for many predators, are relatively abundant, and commonly make broad migration movements. A clearer understanding of the mule deer genome can improve our knowledge of its population genetics, movements, and demographic history, aiding in conservation efforts. Their large population size, continuous distribution, and diversity of habitat make mule deer excellent candidates for population genomics studies; however, few genomic resources are currently available for this species. Here, we sequence and assemble the mule deer genome into a highly contiguous chromosome-length assembly for use in future research using long-read sequencing and Hi-C technologies. We also provide a genome annotation and compare demographic histories of the mule deer and white-tailed deer using the pairwise sequentially Markovian coalescent model. We expect this assembly to be a valuable resource in the continued study and conservation of mule deer.

**Subjects**   Animal and Plant Sciences, Animal Genetics, Functional Genomics

## DATA DESCRIPTION

### Background

The mule deer (*Odocoileus hemionus*; NCBI:txid9872) is a mid-sized ruminant (50–90 kg; Figure 1) [1, 2], ranging from the Yukon Territory in Canada to central Mexico. Mule deer can be found in boreal forests, high- and low-elevation desert shrublands, subalpine forests, woodlands, prairies, and various other habitats, with subspecies and types frequently

**Figure 1.** Photograph of *Odocoileus hemionus*. Photo courtesy of the Utah Division of Wildlife Resources.

inhabiting different habitats [3]. They belong to the Cervidae family, one of the most speciose families in the mammal suborder Ruminantia [4]. Eleven subspecies of mule deer have been recognized, but these are grouped into two morphologically distinct types: mule deer (*O. h. hemionus, fulginatus, californicus, inyoensis, eremicus, crooki, peninsulae, sheldoni,* and *cerrosensis*) and black-tailed deer (*O.h. columbianus*, and *sitkensis*) [5]. While the two types are well-supported by morphological and DNA evidence, little divergence has been observed among the subspecies within each type [6, 7]. This is probably caused by large population sizes and the frequency of long-distance dispersal by individual deer maintaining gene flow among populations [8, 9].

Characteristics such as large population size, diversity of habitat and capacity for long-distance dispersal make mule deer a good candidate species for genomic study [10–12]. However, limited genomic resources are available for *Odocoileus* spp. and include primarily various microsatellite loci [13–15] and molecular resources gleaned from the bovine genome [16–18]. Recently, Russell et al. published the first draft whole genome sequence assembly and a species-diagnostic single nucleotide polymorphism (SNP) panel specifically for mule deer [19]. However, this assembly was based on low-coverage short-read sequencing (Illumina) and was assembled using a reference-based approach, limiting identification of large structural variants. In addition, a short-read based assembly for *O. hemionus sitkensis* has been published in the National Center for Biotechnology (NCBI) database (Bioproject PRJNA476345); however, it is low in contiguity and includes a small number of expected universal single-copy orthologs (Table 1).

## Context

Here, we report a high-quality, chromosome-length reference genome for mule deer assembled from a combination of long-read (Pacific Biosciences [PacBio]) and short-read (Illumina) sequence data and scaffolded using high-throughput chromosome conformation capture (Hi-C). Our goal was to develop whole genome resources that will help us to better

**Table 1.** Metrics of *Odocoileus hemionus hemionus* and *O. h. sitkensis* genome assemblies.

| Species | *Odocoileus hemionus hemionus* (present study) | *Odocoileus hemionus hemionus* | *Odocoileus hemionus sitkensis* |
|---|---|---|---|
| Accession | PRJNA752226 | PRJNA417092 | PRJNA476345 |
| Sequencing platform (Coverage, ×) | PacBio + Illumina (91.2× + 33.4×) | Illumina (25×) | Illumina (26×) |
| Assembly Length (bp) | 2,609,333,024 | 2,342,543,531 | 2,324,921,134 |
| Contig N50 (Kbp) | 28,570.9 | 65.9 | 9.5 |
| Contig L50 | 25 | 9,382 | 70,477 |
| Scaffold N50 (Kbp) | 72,140.9 | 838.8 | 9.7 |
| Scaffold L50 | 13 | 750 | 69,435 |
| Scaffold N90 (Kbp) | 40,772 | 203.6 | 3.2 |
| Scaffold L90 | 33 | 2902 | 238,619 |
| BUSCO complete (%) | 94.50 | 93.60 | 36.10 |

*BUSCO: Benchmarking universal single-copy orthologs; Kbp: kilobase pairs.

understand questions related to mating systems, parentage assignment, relatedness, estimation of demographic parameters, population genetic analysis, and assessment of population viability [20]. We compare our assembly to other chromosome-length assemblies for the red deer and the cow and find high levels of synteny. We also provide an annotation and estimate demographic histories of both the white-tailed and mule deer using the pairwise sequentially Markovian coalescent (PSMC) model. We discuss how this new genome assembly can be applied to conservation and management of mule deer.

## METHODS

### Sample collection and DNA preparation

A tongue biopsy was collected within 2 hours postmortem from a single female mule deer (*O. h. hemionus*), which was removed for depredation purposes from Woodland Hills, Utah, USA (40°00′ N 111°38′ W, Figure 2). Woodland Hills is located at the southern end of the Wasatch Mountain range at an elevation of 1615 m. Vegetation in the area is dominated by Gambel's oak (*Quercus gambelii*), bigtooth maple (*Acer grandidentatum*), bitterbrush (*Purshia tridentata*), and sagebrush (*Artemisia* spp.). The biopsy sampled was immediately stored on ice and frozen at −80 °C within 12 hours of collection. The sample remained frozen at −80 °C until DNA extraction and sequencing were performed. Genomic DNA was extracted from the tongue tissue after treatment with proteinase K using a Qiagen Genomic Tip Kit for High Molecular Weight DNA following the manufacturer's extraction protocol (Qiagen, Valencia, CA, USA). After extraction, the DNA was visualized with pulsed-field gel electrophoresis to evaluate whether the DNA was of sufficient length for single-molecule real-time (SMRT) sequencing using the PacBio Sequel II sequencing instrument [21].

### Sequencing and assembly

The DNA extractions were successful on the first attempt and the pulsed-field gel showed sufficient DNA length, with a band above 50 kilobase pairs (Kbp). The extracted DNA was sheared to 65 Kbp and then size-selected for fragments greater than 32 Kbp using a BluePippin (Sage Science, USA). The size-selected DNA was prepared into a PacBio library using the SMRTbell® Express Template Preparation Kit 2.0 (Pacific Biosciences, USA), then sequenced across two PacBio Sequel II 8M SMRT cells (PN:101-389-001). Each run was performed at the Brigham Young University DNA Sequencing Center (Provo, Utah, USA).

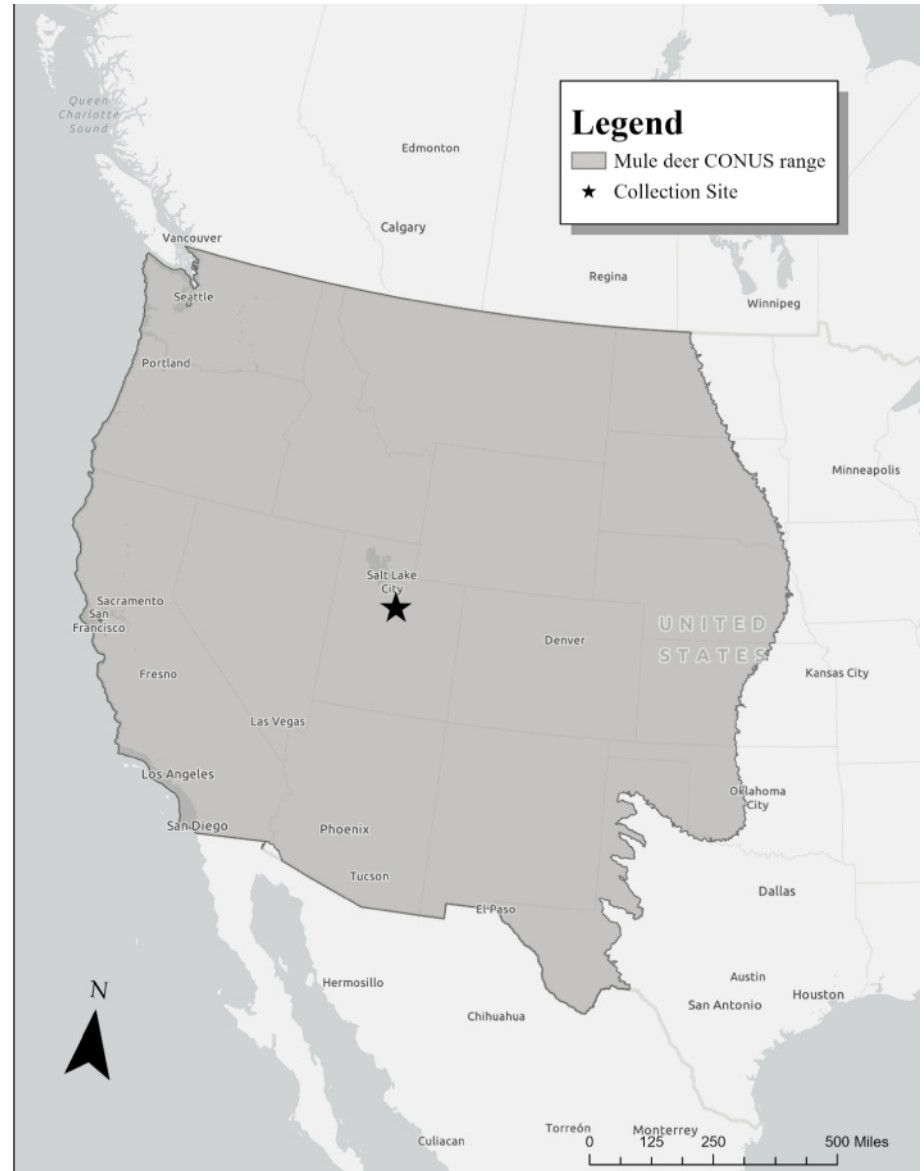

**Figure 2.** Mule deer (*O. h. hemionus*) range in the contiguous USA. Samples for genomic sequencing were collected from Woodland Hills, located in north central Utah, USA. CONUS: Continental United States (of America).

Extracted DNA was also prepared into a paired-end Illumina library with a fragment size of 500 bp. The library was prepared using the NEBNext® Ultra™ II DNA Library Prep Kit for Illumina, and the manufacturer's instructions were followed as outlined in the kit manual (New England BioLabs, Inc., USA). The library was sequenced across two Illumina HiSeq 2500 lanes with 2 × 150-bp paired-end sequencing at the Brigham Young University DNA Sequencing Center (Provo, Utah, USA).

We converted the raw PacBio subreads BAM file to FASTQ using SAMTOOLS v.1.9 (RRID:SCR_002105) [22] and generated a first assembly using WTDBG2 v.2.5-1 (RRID:SCR_017225) with the command parameters "-x sq -g 2.3G -t 80 -L5000." [23]. Reads shorter than 5000 bp were removed and not used in the assembly using the "-L5000"

parameter in WTDBG2. The approximate genome size was estimated using the genome length of the previous mule deer genome and was set to 2.3 gigabase pairs (Gbp). The consensus sequence was derived using the command "wtpoa-cns -t 80 -i <lay.gz file> -fo <output.fa>" [24].

We recovered 239 Gbp of raw PacBio subread data (~90× coverage) from the two PacBio Sequel II SMRT cells. The first SMRT cell generated 114.19 Gbp of subread data with a mean polymerase read length of 23,861 bp and a read N50 of 31,007 bp. The second SMRT cell generated 125.82 Gbp of subread data with a mean polymerase read length of 29,002 bp and a read N50 of 46,596 bp. The Illumina sequencing run yielded ~690 million reads equaling 87.2 Gbp of raw sequence data.

## Genome polishing

We performed an initial error correction step by remapping the PacBio long reads back to the WTDBG2 contig assembly sequence using Minimap2 v.2.17–r941 (RRID:SCR_018550) " -ax map-pb -t 40" and sorting, indexing, and converting the alignment file with the command "sort -o -T reads.tmp" and "index reads.sorted.bam" in SAMTOOLS v.1.9 into BAM format. We performed two rounds of Racon (RRID:SCR_017642) error correction using "-u -t 80" parameters with the PacBio reads, with a separate alignment file created for each run.

We conducted genome polishing with high-fidelity short-read data by first mapping Illumina reads to the Racon-corrected consensus assembly. We first trimmed adapters from the Illumina sequences using Trim Galore v.0.6.4 (RRID:SCR_011847). We then mapped Illumina reads to the Racon corrected assembly using BWA v.0.7.17-r1188 (RRID:SCR_010910) and sorted and indexed the alignment file with SAMTOOLS v.1.9. We used Pilon v.1.23 (RRID:SCR_014731) to correct indel errors using "–vcf –tracks –fix indels –diploid" parameters. We then ran a second round of indel correction by repeating the steps above on the output from the first round of Pilon.

We generated assembly statistics using GAAS Toolkit 1.2.0 and the assembly_stats script [25, 26]. We used BUSCO v.5.2.2 (RRID:SCR_015008) to evaluate the recovery of universal single copy orthologs using the mammalia_odb10 ortholog set [27]. The assembled mule deer genome has a total length of 2.61 Gbp contained in 6033 contigs with a GC content of 41.8%. The contig N50 was 28.6 megabase pairs (Mbp) with a longest contig of roughly 96.5 Mbp. The contig N50 of the new *O. hemionus hemionus* assembly was vastly more contiguous than both the previously published short-read based reference-guided assembly of *O. hemionus hemionus*, which had a contig N50 of 65.9 Kbp, and the short-read-based assembly of *O. hemionus sitkensis*, which had a contig N50 of 9.5 Kbp (Table 1) [19].

## Chromosome-length scaffolding

High-throughput chromosome conformation capture (Hi-C) was performed to provide chromosome-length scaffolding for the consensus genome (Figure 3). Data generation and Hi-C scaffolding was performed by the DNA Zoo Consortium [28]. In brief, in situ Hi-C data [29] was aligned to a draft genome assembly using the Juicer pipeline [30]. The 3D-DNA pipeline [31] was used to error-correct, anchor, order and orient the pieces in the draft assembly, producing a candidate assembly. The candidate assembly was manually reviewed and polished using Juicebox Assembly Tools (JBAT, RRID:SCR_021172) [30, 32]. Interactive contact maps visualized using Juicer.js [33] for before and after the Hi-C scaffolding are available [34] (Figure 4).

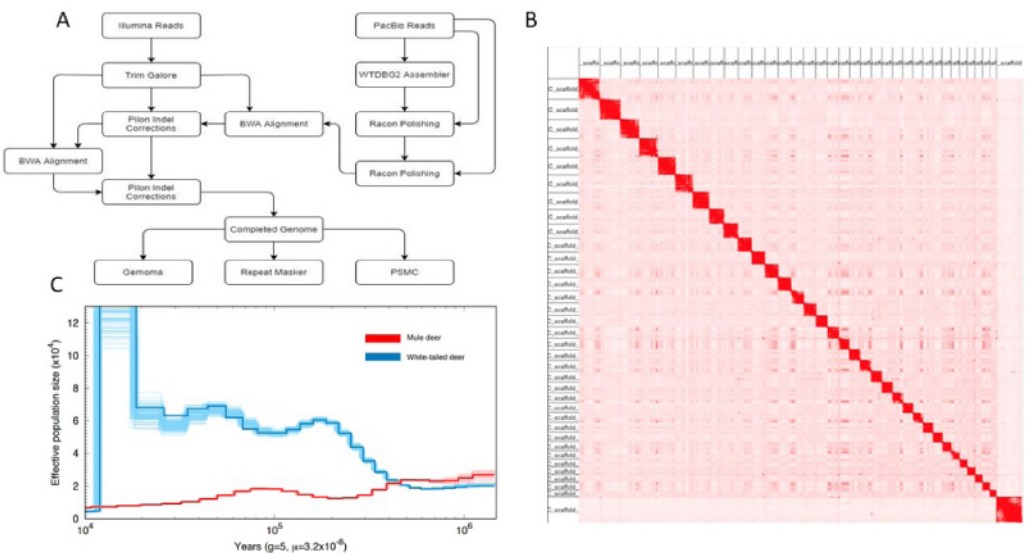

**Figure 3.** Methods flow chart. (A) Summary flow chart of software used for the genome assembly with wtdbg2 and annotation with GeMoMa of *Odocoileus hemionus*. (B) Hi-C contact map of the 35 chromosome-length scaffolds. 93.45% of the genome is held in these chromosome-length scaffolds. (C) Demographic histories estimated with PSMC for *Odocoileus virginianus* and *O. hemionus*.

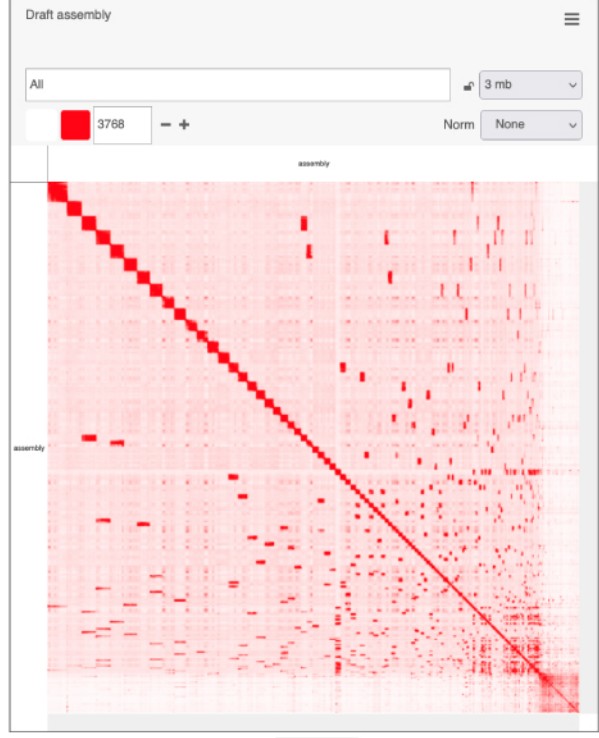

**Figure 4A.** Interactive contact map visualized using Juicer.js for the draft assembly.
https://www.dnazoo.org/assemblies/Odocoileus_hemionus

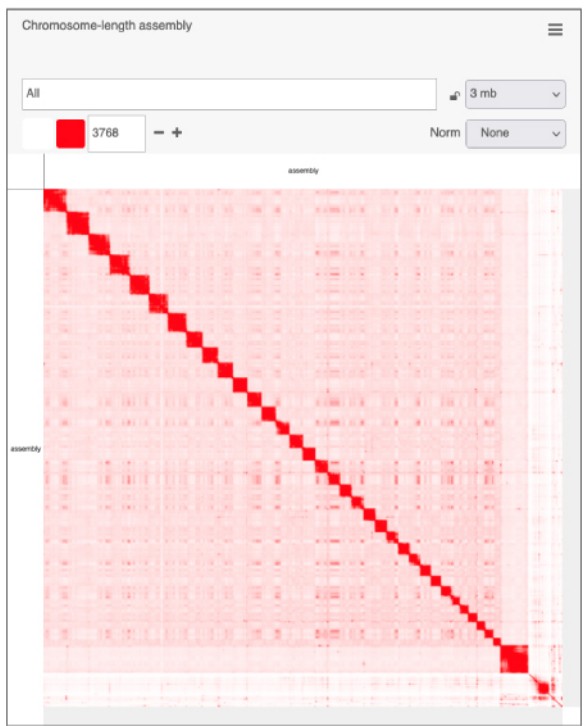

**Figure 4B.** Interactive contact map of the chromosome-length assembly with Hi-C scaffolding. https://www.dnazoo.org/assemblies/Odocoileus_hemionus

The Hi-C scaffolding placed 93.45% of the total basepairs in the assembly into 35 chromosome-length scaffolds, consistent with the known chromosome number, $2n = 70$ [35]. The contiguity also improved, with a scaffold N50 of 72.1 Mbp and L50 of 13. The final scaffolded assembly included 5510 scaffolds and 256,100 $N$s. We successfully identified 94.5% of BUSCO genes in the assembly, with 91.2% single copy and 3.3% duplicated BUSCOs, comparable with other recently published cervid genomes (Table 1) [36]. This also represents an improvement to the previous *O. h. hemionus* assembly, which had a BUSCO score of 93.6%, and a vast improvement to the *O. hemionus sitkensis* genome, which had a BUSCO score of 36.1%.

## Synteny analysis

We aimed to establish syntenic relationships between *O. hemionus* and previously published cervid genomes (Figure 5). We therefore downloaded the *Cervus elaphus* assembly ($2n = 68$, mCerEla1.1, GenBank accession GCA_910594005.1) [37] and the *Bos taurus* assembly ($2n = 60$, ARS-UCD1.2, GenBank accession GCA_002263795.2), and aligned each to the largest 35 scaffolds of our *O. hemionus* ($2n = 70$) assembly. The alignment was carried out with nucmer v4.0 (RRID:SCR_018171) [38], using the command "nucmer -t 24 -c 1000 <REF.fa> <QUERY.fa>" to restrict alignments to those greater than 1 Kbp in length. We extracted the alignment coordinates using the command "show-coords -c -l -L 1000 -T <ALIGN.delta> <COORDS.txt>" from the nucmer package, and visualized the mappings with custom R scripts (R v 4.0.3) [39] using the packages circlize (RRID:SCR_002141) [40] and ggplot2 (RRID:SCR_014601) [41]. Visualizations show alignments longer than 50 Kbp

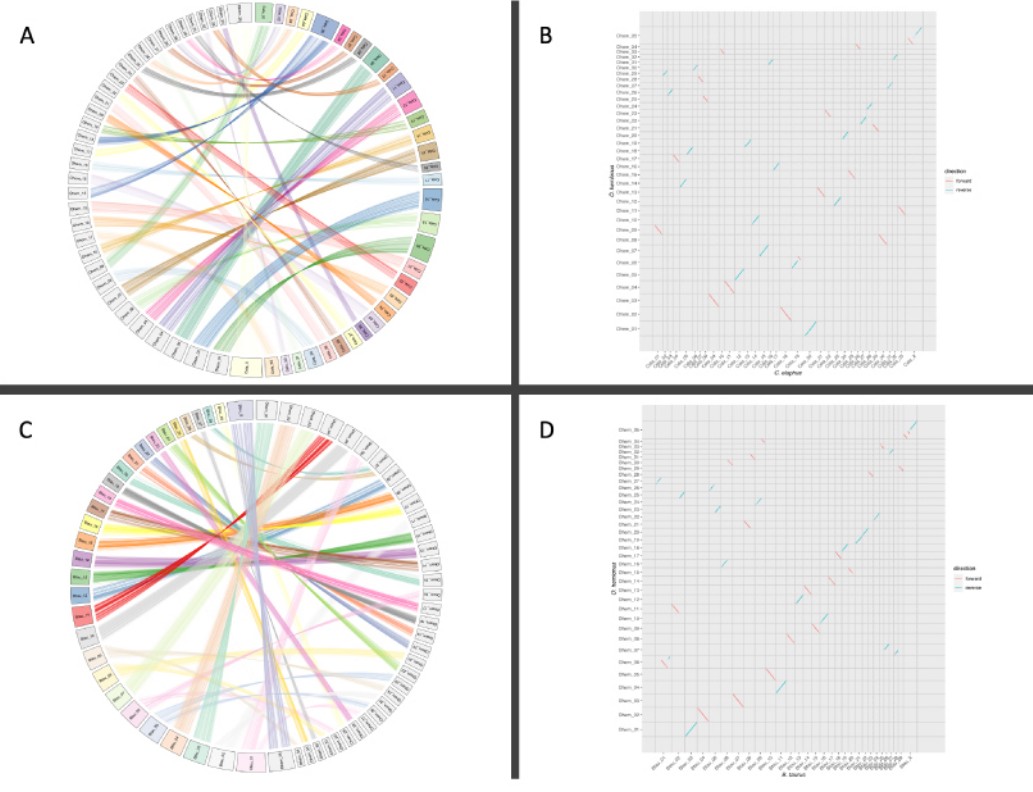

**Figure 5.** Synteny plots of mule deer vs. roe deer and cow. (A) Synteny cord plot of 50-Kbp alignments on chromosome-length scaffolds between *Odocoileus hemionus* and *Cervus elaphus*. (B) Dot plot of 50-Kbp alignments on chromosome-length scaffolds between *O. hemionus* and *C. elaphus*. (C) Synteny cord plot of 10-Kbp alignments on chromosome-length scaffolds between *O. hemionus* and *Bos taurus*. (D) Dot plot of 10-Kbp alignments on chromosome-length scaffolds between *O. hemionus* and *B. taurus*.

(*C. elaphus*) or 10 Kbp (*B. taurus*) to simplify plots. Syntenic chromosomes were identified as those that shared the most aligned bases. There were no ambiguous cases.

## Genome annotation

RepeatMasker (RRID:SCR_012954) [42] was used with the NCBI engine to estimate the overall repeat content of the genome [43]. Repeat databases were built using RepeatModeler v.2.0.1 (RRID:SCR_015027) with parameters "BuildDatabase -name -engine ncbi && RepeatModeler -engine ncbi -pa 8 -database". RepeatMasker v.4.1.1 (RRID:SCR_012954) was used to identify repeats using the parameters "–pa 16 –gff –nolow –lib" (Table 2).

We performed homology-based gene prediction using Gene Model Mapper (GeMoMa) v.1.6.4 (RRID:SCR_017646) with the existing *Odocoileus virginianus* [44] genome annotation used as a reference; the following command was used: "GeMoMa -Xmx50G GeMoMaPipeline threads=40 outdir=annotation_out GeMoMa.Score=ReAlign AnnotationFinalizer.r=NO o=true t=mule_deer.fa i=white_tail a=GCF_002102435.1_Ovir.te_1.0_genomic.gff g=GCF_002102435.1_Ovir.te_1.0_genomic.fna". The GeMoMa annotation predicted 21,983 full-length proteins. Only annotations of coding sequences (CDS) were performed; 5′ or 3′ untranslated regions (UTRs) were not annotated.

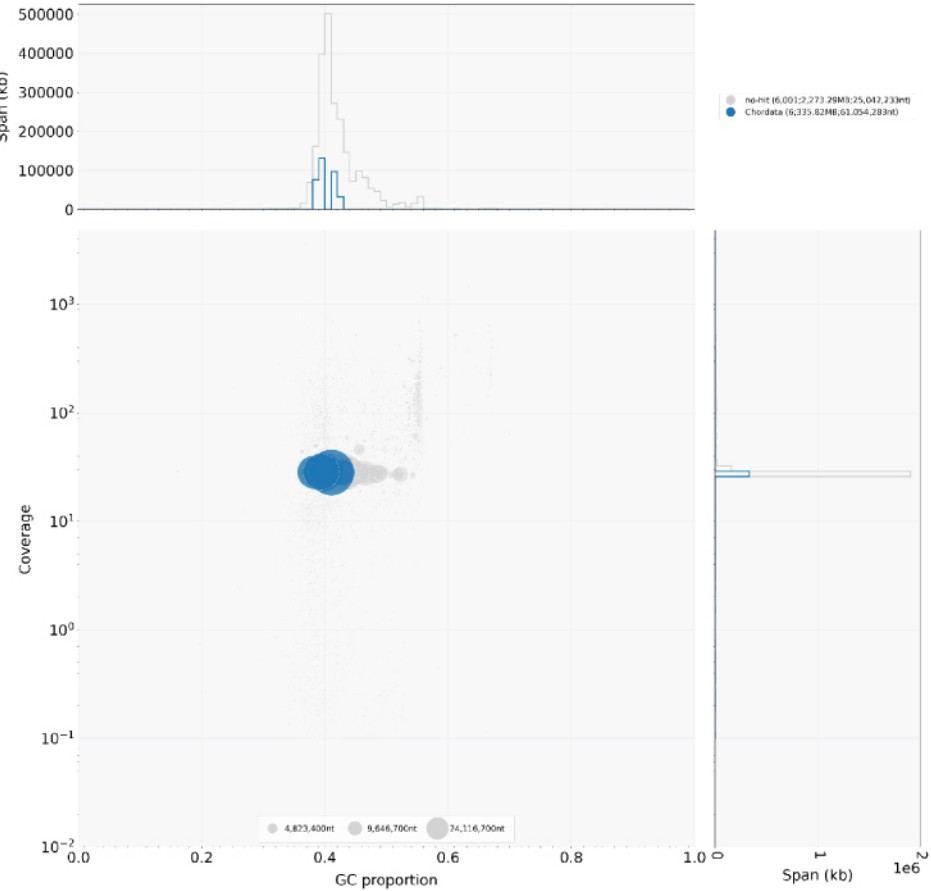

**Figure 6.** Blobplot of the mule deer genome assembly. All contigs are assigned to Chordata.

| Table 2. RepeatMasker annotation. | |
|---|---|
| **Annotation type** | **Percentage (%)** |
| Retroelements | 31.72 |
| DNA transposons | 1.38 |
| Rolling circles | 1.62 |
| Interspersed repeats | 37.86 |
| Small RNA | 0.80 |
| Satellites | 0.95 |
| Simple repeats | 0.04 |
| Unclassified | 4.76 |

Blobtools v1.1.1 (RRID:SCR_017618) was used to evaluate the assembled genome for possible contamination. BLAST v.2.9.0 (RRID:SCR_004870) was used to identify any possible contamination using the command "blastn -task megablast -outfmt '6 qseqid staxids bitscore std' -max_target_seqs 1 -max_hsps 1 -num_threads 16 -evalue 1e-25". A blobplot was created for visualization using the "create" function of blobtools. The blobplot revealed no evidence of contamination in the genomes (Figure 6).

## Historical demography

We used pairwise sequentially Markovian coalescent (PSMC) v.0.6.5-r67 (RRID:SCR_017229) to estimate the demographic history of the mule deer and the previously sequenced white-tailed deer [45]. We realigned Illumina reads to the final assembly with BWA and sorted and indexed the alignment file in SAMTOOLS v.1.9. Average coverage for the white-tailed deer and mule deer were 19× and 33×, respectively. We used mpileup and bcftools to call heterozygous sites using the command "samtools mpileup -C50 -uf" and "bcftools call -c", respectively. Additionally, Bcftools v.1.11 (RRID:SCR_005227) was used with the vcfutils.pl utility and the following parameters "vcf2fq -d 10 -D 90". We then used PSMC v.0.6.5-r67 to generate the demography history. We first created a psmcfa file with the following command "fq2psmcfa -q20". The psmcfa file was split using the "splitfa" function of PSMC. A PSMC was created using the command "psmc -N25 -t15 -r5 -p '4+25*2+4+6". Bootstraps were created from the split psmcfa file using the command "seq 100 | xargs -i echo psmc -N25 -t15 -r5 -b -p "4+25*2+4+6" -o round-[46].psmc $splitpsmcfa | sh". The initial PSMC and bootstraps were then merged and visualized with psmc_plot.pl using the command "psmc_plot.pl -pY20 -g5 -u 3.22e-8". A generation length of 5 years was used based on the roe deer (*Capreolus capreolus*) generation time [47].

To compare demographic histories with the other most common North American deer species, the white-tailed deer (*Odocoileus virginianus*), we followed the same process described above. We downloaded the *O. virginianus* assembly from NCBI (accession: NC_015247) and downloaded the raw Illumina reads from the sequence read archive (SRA) using the fastq-dump, utility within SRAtoolkit v.2.10.9, with the following parameters "fastq-dump –gzip –skip-technical –readids –read-filter pass –dumpbase –split-e –clip". Because fastq-dump alters read names, individual read names were corrected to match in both the forward and reverse fastq files by removing ".1" from the end of the forward reverse identifier and ".2" from the end of the reverse sequence identifier.

We used PSMC analysis to compare historic population trends of *O. hemionus* and *O. virginianus*. In doing so, we observed that *O. hemionus* and *O. virginianus* have divergent demographic histories. As effective population size for *O. hemionus* increases, the effective population size of *O. virginianus* appears to decrease, and vice versa. The effective population size of *O. hemionus* has been in constant decline since the most recent glacial period roughly 500,000 years ago. Two possible explanations for this decline are overall population decline or population fragmentation. This pattern is divergent from *O. virginianus*, which has shown increases in effective population size since the same time period. While both deer species inhabit the same continent, and even possess some overlapping habitat, it appears that the species react differently to environmental changes (Figure 3). However, without denser sampling of the white-tailed deer, we cannot rule out the possibility that this pattern of increase in effective population size has emerged because of some recent migration or admixture event with a more distantly related population.

## RE-USE POTENTIAL

Our high-quality draft genome of the mule deer represents an advance in available genomic data for *Odocoileus*. With a total length of 2.6 Gbp and a contig N50 of 28.6 Mbp, this chromosomal-length *de novo* assembly can serve as a base for future conservation and genomics research. The importance of deer at the ecosystem level and to local economies means that continued efforts to conserve these populations are vital [48]. Our hope is that



this genomic resource will further our understanding of *O. hemionus*, and subsequently lead to more effective management of the species, including insights into the impact of anthropogenic barriers on gene flow, the possibility of species divergence in isolated populations, and the presence of multiple paternity [49, 50].

## DATA AVAILABILITY

The *Odocoileus hemionus* genome and raw reads are publicly accessible through NCBI. The genome data is available via BioProject ID: PRJNA752226. The raw Hi-C data is available via PRJNA512907. The data sets supporting the results of this article are available in the *GigaScience* Database [51].

## DECLARATIONS
## LIST OF ABBREVIATIONS

bp: base pair; BUSCO: Benchmarking Universal Single-Copy Orthologs; Gbp: gigabase pair; Hi-C: high-throughput chromosome conformation capture; Kbp: kilobase pair; Mbp: megabase pair; NCBI: National Center for Biotechnology Information; PacBio: Pacific Biosciences; PSMC: pairwise sequentially Markovian coalescent; SMRT: single-molecule real-time.

## COMPETING INTERESTS

The authors declare that they have no competing interests.

## FUNDING

This work was supported by the following funding sources: State of Utah, Utah Division of Wildlife Resources (DWR) Research Grant (grant number MG19163SS) to B.R.M.; BYU Life Sciences College Undergraduate Research Award to A.M.T., National Science Foundation Physics Frontiers Center Award (grant number PHY1427654), Welch Foundation (grant number Q-1866), US Department of Agriculture (USDA) Agriculture and Food Research Initiative Grant (number 2017-05741), and National Institutes for Health (NIH) Encyclopedia of DNA Elements Mapping Center Award (grant number UM1HG009375) to E.L.A. The DNA Zoo sequencing effort is supported by Illumina, Inc.; IBM; and the Pawsey Supercomputing Center.

## AUTHOR CONTRIBUTIONS

S.L., P.B.F., R.T.L., and B.R.M. designed this project. A.M.T. prepared the samples. A.M.T., N.B.E., and P.B.F. performed the analyses. R.K., D.W., O.D. and E.L.A. generated the Hi-C data and performed the scaffolding. S.L., A.M.T., T.A.H., P.B.F., B.R.M., R.T.L., N.B.E., and P.B.F. wrote and revised the manuscript. All authors approved the final version of the manuscript.

## ACKNOWLEDGEMENTS

We thank Seth B. Wilson and Ethan Tolman for assistance with our GenBank upload and PSMC. We thank Edward Wilcox from the BYU Sequencing Center for assistance with DNA sequencing. All computation was performed on the Fulton Supercomputer (Brigham Young University, Provo, Utah, USA). Hi-C data for the mule deer were created by the DNA Zoo Consortium.

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
