## [Reviewer Report]

Comments on revised manuscriptThank you for the authors for revising the manuscript according to the reviewers' requests and congratulations for this excellent work! It is now acceptable without any further changes.

---

## [Reviewer Report]

Reviewer name and names of any other individual's who aided in reviewer Endre BartaDo you understand and agree to our policy of having open and named reviews, and having your review included with the published papers. (If no, please inform the editor that you cannot review this manuscript.)YesIs the language of sufficient quality?YesPlease add additional comments on language quality to clarify if needed
Are all data available and do they match the descriptions in the paper? YesAdditional CommentsAre the data and metadata consistent with relevant minimum information or reporting standards? See GigaDB checklists for examples <a href="http://gigadb.org/site/guide" target="_blank">http://gigadb.org/site/guide</a>YesAdditional CommentsIs the data acquisition clear, complete and methodologically sound?YesAdditional CommentsIs there sufficient detail in the methods and data-processing steps to allow reproduction?YesAdditional CommentsIs there sufficient data validation and statistical analyses of data quality? YesAdditional CommentsIs the validation suitable for this type of data?YesAdditional CommentsIs there sufficient information for others to reuse this dataset or integrate it with other data?YesAdditional CommentsAny Additional Overall Comments to the AuthorThis manuscript by Lamb et al describes the denovo genome sequence assembly of the mule deer. This is a precisely carried out work, which used the standard programs and pipelines. Most importantly, the isolated DNA was suitable for the Sequel II sequencing and therefore it resulted in very long contigs. The scaffolding were done at the DNA Zoo consortium with the help of Hi-C data. The annotation was based on the already available annotation of the close relative Odocoileus virginianus.
In summary, the resulted reference genome sequence is suitable for further analysis and will be a good basis for the incoming population genomics, transcriptomics and comparative genomics works.

I have only a few minor comments and questions:

1. The authors speak about "Chromosome-length Scaffolding" but there was no effort to put the scaffolds into chromosomes (there are several chromosome -level assemblies available from Cervidae and of course the bovine assembly could be used as well). As it seems that most of the Cervidae have 2n=70 this could be a reliable approach
2. The authors used PSMC for estimating the demographic history of the mule deer but there is no word about the number of heterozygous sites. The available other short read WGS could have been used for this as well.
3. There is no comparison with the other (although only short read based) mule deer and Cervidae assemblies
4. The numbers of contigs and scaffolds and the Ns would be mentioned in the text
5. The authors used the -g 2.3G parameter at the assembly but finally got 2.6 Gbp as the genome size. Did they try to re-run the analysis with the -g 2.6G parameter?
6. The author should mention that the annotation is only cds based (there are no 5' and 3' UTRs annotated)
7. The authors should refer to Figure 3 instead of Figure 4 at the Blobtools paragraph
8. The resolution of the Figures 3,4 is very low
9. What is "genoma" in Figure 2A?RecommendationMinor Revision

---

## [Reviewer Report]

Reviewer name and names of any other individual's who aided in reviewer Rebecca S. TaylorDo you understand and agree to our policy of having open and named reviews, and having your review included with the published papers. (If no, please inform the editor that you cannot review this manuscript.)YesIs the language of sufficient quality?YesPlease add additional comments on language quality to clarify if needed
Are all data available and do they match the descriptions in the paper? YesAdditional CommentsI checked the two links included in text as well as the NCBI data availability and all data is available for download with good explanations as to what all the files are.Are the data and metadata consistent with relevant minimum information or reporting standards? See GigaDB checklists for examples <a href="http://gigadb.org/site/guide" target="_blank">http://gigadb.org/site/guide</a>NoAdditional CommentsOn the whole the information included for the data and metadata is good, but perhaps some more information about the sample used would be beneficial. It is stated that the sample came from 'Woodland Hills, Utah'. I assume this is in the United States? Some more information about the environment would be beneficial for those unfamiliar with the area. It is also not stated which subspecies the sample belongs to. A map of the species range and where this sample is from could also help. 

Additionally, in the 'Background and context' section, you state that 'genetic resources available for Odocoileus spp. are limited to a variety of microsatellite loci' with the exception from Russell et al. I think you need to do a more thorough search – for example I have used a sitka deer genome (Odocoileus hemionus sitkensis) as an outgroup for my work, sequenced by the CanSeq150 program, found on the NCBI under Bioproject PRJNA476345.Is the data acquisition clear, complete and methodologically sound?NoAdditional CommentsI was a bit confused by the sentence 'The assembled mule deer genome has a total length of 2.61 Gbp with a GC content of 41.8% and a contig N50 of 28.6 Mbp (Table 1) with a longest contig of roughly 96.5 Mbs' occurring before the 'Chromosome-length Scaffolding' section. Are these assembly statistics for the version of the genome before chromosome scaffolding? It would be better to report the final assembly statistics for the chromosome scale assembly, or if these are the final statistics then move those results until after the 'Chromosome-length Scaffolding' section. 

For Table 1, it might be nice to include the L/N90. The most standard are the N/L 50 and the N/L90, so you don’t necessarily need all of the others. 

Additionally, I could not find where it is stated how many 'chromosomes' in the final assembly. Is this a chromosome assembly or (more likely) a chromosome scale assembly (so there are also other scaffolds not included in the main ‘chromosomes’)? Relevant paper newly published might be good to reference: Yamaguchi et al. Technical considerations in Hi-C scaffolding and evaluation of chromosome-scale genome assemblies. Molecular Ecology.

I also think it would be beneficial to know what the coverage of the files you used for the PSMC analysis were, making sure to filter out ~double the average (I can see that you filtered for a maximum of 90X but I don't know whether this is appropriate or not). Also a citation for the generation time used would be beneficial as this strongly influences PSMC results. It would also be good to explain the rise in effective pop size seen recently in the white tailed deer. Is this a real pattern or because PSMC can be spurious at more recent times? This is also a reason why it would be good to know the depth of both files used here.

Could the different demographic histories be caused by competition between the species? I am not an expert on these species but I was just curious given their contrasting demographic histories.Is there sufficient detail in the methods and data-processing steps to allow reproduction?YesAdditional CommentsIs there sufficient data validation and statistical analyses of data quality? Not my area of expertiseAdditional CommentsIs the validation suitable for this type of data?YesAdditional CommentsIs there sufficient information for others to reuse this dataset or integrate it with other data?NoAdditional CommentsAs I stated above, information about which subspecies this individual is from in text would be beneficial. Any Additional Overall Comments to the AuthorI do believe this to be a high quality chromosome scale assembly which will be beneficial for research of the species, as well as for comparative analyses between species.RecommendationMinor Revision